# Health systems analysis and evaluation of the barriers to availability, utilisation and readiness of sexual and reproductive health services in COVID-19-affected areas: a WHO mixed-methods study protocol

Seni Kouanda,[1] Eunice Nahyuha Chomi,[1] Caron Kim [iD],[2] Sothornwit Jen,[3] Luis Bahamondes [iD],[4] Jose Guilherme Cecatti [iD],[5] Pisake Lumbiganon,[3] Modey Emefa,[6] Vanessa Brizuela [iD],[2] Hamsadvani Kuganantham,[2] Armando Humberto Seuc,[2] Moazzam Ali [iD],[2] WHO HRP Social Science Research Team

For numbered affiliations see end of article.

**Correspondence to**
Dr Moazzam Ali;
alimoa@who.int

## ABSTRACT

**Introduction** COVID-19 has led to an unprecedented increase in demand on health systems to care for people infected, necessitating the allocation of significant resources, especially medical resources, towards the response. This, compounded by the restrictions on movement instituted may have led to disruptions in the provision of essential services, including sexual and reproductive health (SRH) services. This study aims to assess the availability of contraception, comprehensive abortion care, sexually transmitted infection prevention and treatment and sexual and gender-based violence care and support services in local health facilities during COVID-19 pandemic. This is a standardised generic protocol designed for use across different global settings.

**Methods and analysis** This study adopts both quantitative and qualitative methods to assess health facilities' SRH service availability and readiness, and clients' and providers' perceptions of the availability and readiness of these services in COVID-19-affected areas. The study has two levels: (1) perceptions of clients (and the partners) and healthcare providers, using qualitative methods, and (2) assessment of infrastructure availability and readiness to provide SRH services through reviews, facility service statistics for clients and a qualitative survey for healthcare provider perspectives. The health system assessment will use a cross-sectional panel survey design with two data collection points to capture changes in SRH services availability as a result of the COVID-19 epidemic. Data will be collected using focus group discussions, in-depth interviews and a health facility assessment survey.

**Ethics and dissemination** Ethical approval for this study was obtained from the WHO Scientific and Ethics Review Committee (protocol ID CERC.0103). Each study site is required to obtain the necessary ethical and regulatory approvals that are required in each specific country.

## STRENGTHS AND LIMITATIONS OF THIS STUDY

⇒ The study approach and design enable a comprehensive analysis of the barriers to availability, utilisation and readiness of sexual and reproductive health services during COVID-19 as well as the postpandemic recovery as transmission is contained.

⇒ The use of a mainly qualitative approach places women's rights and needs during health emergencies at the centre of the debate, underscoring the need for more responsive policies.

⇒ The inclusion of partners will enhance understanding of gender dynamics and support efforts towards identifying strategies to enhance positive male involvement and engagement in women's sexual reproductive health and rights.

⇒ A possible limitation is that the study will be conducted during and after one of the waves of the pandemic, at which stage there may have been changes in the health system as lessons are learnt.

⇒ Another limitation is that the estimation of the recovery period may be difficult in settings where the pandemic is yet to be brought under control.

⇒ This part of the analysis may be subject to delays until an appropriate period is determined.

## INTRODUCTION

The COVID-19 disease, which was first reported in December 2019 and rapidly spread globally, has led to an unprecedented increase in demand on health systems to care for people infected, necessitating the allocation of significant resources, especially medical resources, towards the response. The increased demand, compounded by the restrictions on movement instituted as part

of containment measures may have led to disruptions in the provision of essential services, including sexual and reproductive health (SRH) services.[1 2]

COVID-19 is new to humans and only limited scientific evidence is available on the impact of COVID-19 on SRH service delivery. However, lessons from the Ebola and Zika virus outbreaks have highlighted the severe disruptions in SRH services that expose women and girls in particular to preventable health risks.[2 3] Some services may be unavailable due to either facilities and health workers being repurposed to care for patients with COVID-19, patient safety concerns, movement restrictions disrupting travel to health facilities, supply chain disruptions or a reduction in health workers because of increasing numbers being themselves infected by COVID-19.[4 5] In addition, overwhelmed with COVID-19 cases, clinical staff may not have the time or personal protective equipment (PPE) needed to provide family planning counselling and commodities.[5] Recent evidence suggests disruptions lasting 3–6 months in 2020 left between 4 and 23 million women in low-income and middle-income countries unable to access modern contraceptives, a projected 1.4 million (500 000–2.7 million) unintended pregnancies and an additional 31 million cases of sexual and gender-based violence (SGBV).[1 6–9] Furthermore, studies have modelled the potential impact, showing that even a 10% reduction in essential SRH services could lead to an estimated 15 million unintended pregnancies, 3.3 million unsafe abortions and 29 000 additional maternal deaths during the next 12 months.[7–10] Continuity of essential health services while keeping people safe during the response to disease outbreaks such as the COVID-19 pandemic is therefore essential for the prevention of both direct and indirect mortality.[4 5]

This is a generic standardised protocol designed to maximise the likelihood that data are systematically collected and shared rapidly in a format that can be easily reproducible, aggregated, tabulated and analysed across many different settings globally and be useful as templates for use in health emergencies in the future. This will facilitate the comparison of results across regions and countries and will potentially improve the quality of observational studies by identifying and minimising biases. Given that use in different settings will require some adaptation, these possibilities have been highlighted throughout the protocol.

The introduction should be updated with country specific data on COVID-19 epidemiology and current research findings prior to submission to local/country national institutional review boards

The main aim of this study is to assess the availability of contraception, comprehensive abortion care, sexually transmitted infection (STI) prevention and treatment (including HIV) and SGBV care and support services in local health facilities during COVID-19 pandemic. The four specific objectives of the study are:

1. To explore the availability of, and health facility readiness to provide these services in areas most affected by COVID-19.
2. To assess the availability and quality of services and barriers to the utilisation of these services from clients' and providers' perspectives in the selected COVID-19-affected areas.
3. To assess the postpandemic recovery (*postpandemic recovery refers to the period when ideally health facilities have been able to recover from the disruption in service provision, following reduced transmission levels. Caution should be applied to the term 'postpandemic' as services may be still impacted even with lower levels of transmission. Therefore, the research team should be cognizant of the distinct time periods in their facility assessments*) of the facilities in the provision of these services in comparison to the pandemic period.
4. To enhance the SRH service capacity in COVID-19 through advocacy, policy briefs, media dissemination and academic papers towards national and regional stakeholders including policymakers, academia, healthcare providers and the community.

## METHODS AND ANALYSIS
### Study design
This is a repeated cross-sectional study, using both quantitative and qualitative methods. The two data collection points, baseline and end-line, will be 9–12 months apart. The aim of having two data collection points is to document and share the local evidence with authorities on the SRH services at baseline, and then at end line to track and demonstrate the changes and improvement in services over time. The WHO situational-level assessment of COVID-19 transmission will be used as the basis for determining when to conduct baseline and end point data collection[11]:

► Level 0: no known transmission in the preceding 28 days, no restriction on daily activities.
► Level 1: basic measures in place, clusters of cases reported controlled with basic measures, limited and transient disease.
► Level 2: local transmission from imported cases to close contacts. Contact tracing, physical distancing and quarantine measures can contain the spread.
► Level 3: community transmission, where source of infection is untraceable, outbreak rapidly spreads in clusters.
► Level 4: disease outbreak has become an epidemic, where there are major clusters of infection all over the country, high number of deaths and it is very difficult to control transmission without strict containment measures.

Countries should time the baseline and end-line data collection as follows: Baseline data collection is proposed when countries are experiencing level 3 or 4 transmission (referred to here as peak transmission/pandemic period), as disruptions in service provision

are likely to occur during these periods. End-line data collection is proposed when countries have tentatively managed to contain the spread and are at levels 0,1 or 2 (referred to here as post-pandemic period/after the pandemic) after experiencing levels 3 or 4.

Given the dynamic nature of the epidemic, where each country is experiencing varying levels of COVID-19 transmission (and the challenges from the resurgence of variants and sudden increase in the number of cases), it is difficult to assume standardized pandemic conditions at country level. Therefore, the time interval between data collection points is just an estimation. For countries that have contained the spread, it may be decided to collect baseline data retrospectively, selecting the peak period. However, strategies to minimize or avoid recall bias should be used.

## Study setting

The study will be conducted in geographical areas selected based on the epidemic status. To get variation in responses, health facilities will be selected as focal points considering their qualification to provide SRH services and other criteria for the researchers to access community members for participation in the study. It is expected that variation in health facilities distributed within different geographical areas will provide access to communities of all socioeconomic backgrounds (countries included are: Brazil, Burkina Faso, China, England, Ghana, Italy, Kenya, Pakistan, Thailand).

> Due to various local realities and political factors, countries should select the research study sites based on geographical location, organization of SRH service delivery and epidemic status where COVID-19 is likely to have significantly affected service delivery (given that even within countries the transmission status differs, hence the differential impact on facilities in different geographical areas). Consultation with the Ministry of Health will help identify the areas most affected by COVID-19.

## Study population

The study population will be women seeking SRH services from the selected health facilities and the partners, who will be from different households. This technique will offer a measure of protection for those women who may be at risk just because they participated in the study.

> In contexts where it is not customary for men to accompany their partners when they seek SRH services, other options can be used to access them. Healthcare providers can be medical doctors, nurses, midwives, nurse assistants, allied health professionals and other cadres depending on the norms and standards for the provision of SRH services in different country contexts.

## Sample selection

Each sample will be selected based on specific criteria, as well as specific country contextual factors.

### Health facility selection

The minimum criteria for selection should include availability of human resources, primarily the qualification to provide contraception, safe abortion, including the treatment of complications and the provision of postabortion care, STI care and treatment and support for women experiencing SGBV. In addition, the diversity of health facility capacity, administrative rank, urban/rural setting and the willingness of the providers in charge to participate in the study, as well as how the COVID-19 response has been organised in terms of treatment centres. These criteria for selection of health facilities within the geographical sites were made to encourage a representative mix of facilities. The same health facilities will be used for both data collection points to highlight the changes in SRH delivery due to COVID-19 and postpandemic recovery.

> Given the dynamic nature of the pandemic some flexibility should be given to all study sites in the selection of facilities, maintaining the minimum criteria specified above.

### Qualitative sample
#### Selection of women

Women will be recruited for in-depth interviews (IDIs) and focus group discussions (FGDs). The women will be purposively selected to obtain a sample of women and the partners. Selection will be based on (1) being of reproductive age (18–49 years) and (2) having sought or tried to receive SRH services from local health facilities. The study will use gatekeepers, who will be healthcare providers not involved in the study to approach the women as they leave the health facility. Their role will be limited to the identification and introduction of potential participants to the study team.

For the IDIs, a sample of 10~15 women (or until saturation) and 6~12 partners (or until saturation) will be recruited. For countries with more enrolled health facilities in the study, the maximum number of the IDIs will be with 10 women (or until saturation) at reproductive age seeking for reproductive healthcare services, and with 5 partners (or until saturation) per health facility. This will be done as an exit interview and the women will be consecutively selected until the desired sample (or saturation) has been achieved.

For FGDs, gatekeepers will also ask participants about their level of comfort discussing experience seeking SRH issues in a group before being recruited. About six to eight participants per FGD will be recruited, with the expectation of at least two focus groups per facility. Some female and male participants will also be invited for individual interviews after the FGDs.

This is also to note that by interviewing husbands and wives will be from different households; by not

interviewing husband-wife from the same household will operationalise the concept of 'do no harm' and offer a measure of protection for those women who may be at risk just because they participated in the study.

> The ages can be adjusted to account for differences in the nationally recognised reproductive age group in different contexts. Inclusion of girls younger than 18 years should take into consideration ethical issues of assent and consent.

To adjust to challenges posed by COVID-19 transmission, additional participants can be recruited through online recruitment from chat groups such as Facebook, Twitter or WhatsApp. Careful attention must be paid to the ethical issues of privacy and confidentiality of participants and integrity of the researchers (transparency of aims, details, risks and benefits of the study, obtaining the necessary permission to join restricted groups and the presence of the researcher in the group). Other challenges of online recruitment that need attention include retention of participants, potential selection bias, verification of participant identity and comprehension of informed consent.[12] Care needs to be taken by hiding the names and faces (eg, via video conference calls) of the online participants, responsible handling of participants' personal information to minimise the likelihood of embarrassment, loss of dignity or harm because of the online recruitment process to address the online privacy and confidentiality concerns. The moderators will ask the participants to give a short self-introduction, using pseudonyms/numbers, instead of their actual names, and not using video options. This includes not disclosing any of this information without the participants' consent.

### Selection of healthcare providers
Approximately one to two healthcare providers per health facility will be purposively selected and only those who (1) deliver SRH services, (2) are most knowledgeable about readiness and availability of SRH services and (3) are stationed in the SRH clinic and have been working at the clinic for at least 6 months before the pandemic started will be selected for inclusion in the study. This information will be obtained from the facility in-charges.

### Quantitative sample
#### Selection of women
In each health facility, a maximum of 3 clients from each section (postabortion care, family planning, SGBV, STI, abortion) will be consecutively selected to achieve up to 20 clients per health facility.

### Selection of healthcare providers
One healthcare provider, preferably most knowledgeable in the health facility about SRH services provided, as determined by seniority, position or function, will be selected to assist in the health facility assessment.

### Data collection
FGDs and IDIs will be used to understand client's perspectives on their experiences in accessing SRH services during and after peak transmission and healthcare providers' perspectives of SRH service availability and readiness in COVID-19-affected areas. The health system assessment will be used to assess the health system response to COVID-19. Study teams should be aware of the possibility that healthcare provider burnout or heavy workload may affect willingness or ability to participate in the study. Adaptations to data collection should be made to avoid overburdening the healthcare providers and clearly explained during recruitment.

### Data collection tools
There will be five sets of tools for clients: one FGD guide for women (published as online supplemental file 1), one FGD guide for partners/men (published as online supplemental file 2), one interview guide for women (published as online supplemental file 3), one interview guide for men (published as online supplemental file 4) and one interview guide for healthcare providers (published as online supplemental file 5).

For the facility assessment, a facility and readiness assessment questionnaire has been developed based on the following validated tools[13–17]:
1. WHO Service Availability and Readiness Assessment guide.
2. WHO Health Facility Readiness Checklist.
3. WHO Safe Abortion Assessment Tool.
4. SGBV Quality Assurance Tool.

The facility and readiness assessment tool comprises five modules: (1) health services continuation; (2) family planning services; (3) abortion services; (4) STI and (5) SGBV, in line with the study SRH focus areas. The following indicators will be used the assess the availability and readiness:
1. Policies and plans.
2. Service maintenance and referrals.
3. Infrastructure.
4. Commodities.
5. Human resources.

> Each study site will adapt (including translation to local language) the tool to its specific context, taking into consideration the existing policies, SRH treatment guidelines, staffing norms and standards, types of facilities, national essential medicines lists and national health information systems among others. After adaptation to specific country context all tools should be piloted before being used.

### Data collection methods
#### Participant informed consent process
All study participants will be taken through a detailed informed consent process (for participation and audio-recording), which will be documented and those who agree to participate in the study will be asked for signed

written (oral consent for those who cannot write, or for situations that necessitate online data collection) consent.

Given that data collection will be conducted during COVID-19, an assessment of the risks to the study team and participants should be made. The decision of whether to conduct face-to-face or remote data collection should be based on the risk assessment, followed by context-specific recommendations on adaptations to the original data collection process. Remote data collection will necessitate adaptation to the length of interviews and discussions, rapport building, oral informed consent, privacy and confidentiality measures as well as making special arrangements where access to internet and mobile connectivity is limited.[18] Face-to-face data collection will require special arrangements to ensure the safety of the study team and participants, following WHO as well as country-specific safety protocols, including educating and training the research team and participants about COVID-19, provision of PPE and hygiene supplies, mandatory hygiene practices and sanitisation of venues and equipment and physical distancing.[19]

The FGDs and IDIs will be conducted at times and in venues that are considered both convenient and safe for the participants to freely discuss the subject matters. In addition, qualified researchers with experience in conducting IDIs and facilitating FGDs will be trained to ensure the validity of the data collection and will be selected based on the gender of the participants. The FGDs and IDIs will be semi-structured and follow a topic guide specific to each group of participants but will take place as a conversation in which the researchers promote a safe, comfortable environment to enable a comprehensive and candid record. In addition to the semi-structured interview, narrative interview techniques may be used depending on the participants' narratives about life events and reproductive healthcare needs.

The FGDs and IDIs will be audio-recorded. Prior consent for recording will be sought again from each participant. Anonymity of the participants will be ensured by removing any personal or family identifiers and all recorded sessions will be coded for purposes of identification with a date, geographical site and session number. After each session, the recording will be sent to the team supervisor for secure storage for transcription. The audio files will be encrypted and sent to an electronic database which will be shared with WHO. Therefore, only the research team will be authorised to listen to the recordings. These audio recordings will be retained until they have been transcribed and checked for accuracy, after which they will be destroyed.

The FGDs will explore knowledge about COVID-19, care-seeking during COVID-19 and risk perceptions and availability of SRH services (end-line FGD will also explore postpandemic services in comparison with the pandemic period). Before starting the FGD, the facilitators will collect sociodemographic data for each participant, build rapport with and among participants and set ground rules to ensure positive group dynamics that foster an effective and rich discussion. The facilitators will also use this time to know and understand the emic categories (field research and viewpoints obtained from within the social group, from the perspective of the subject) used by locals to describe their perceptions and practices related to the study topics. Each FGD is expected to last approximately 60–90 min.

IDIs with women will explore the psychosocial effects of COVID-19 on fertility desires, knowledge of COVID-19, risk perceptions and concerns about effects on SRH, care seeking behaviours, experience and barriers in accessing SRH services reproductive health, particularly comprehensive abortion, STI prevention and treatment and SGBV care and support and their related needs for accurate information and reproductive health services. IDIs with partners will explore their knowledge of COVID-19, risk perceptions and concerns about effects on SRH, their influence on the access to SRH services of the partners and find out what role they play when the partners require SRH services like contraception, comprehensive abortion care, STI prevention and treatment and SGBV care and support. Each interview is expected to last approximately 40–60 min. IDIs for the women and the partners will be conducted separately.

In some countries, abortion is illegal, posing challenges in data quality since participants, fearing reprisals, may not provide accurate responses to abortion-related questions. Necessary precautions should be taken to ensure the interviews are conducted in a supportive and non-judgemental manner, encouraging the participants to respond freely. These include selection and training of interviewers to enable them to overcome their biases and stereotypes about abortion, building trust and rapport with participants and selection of interview venue to guarantee privacy.[20]

Given the sensitive nature of the data being collected, care must be taken in the selection of data collectors to ensure protection of participants. The gender, experience and attitudes of the data collectors are important considerations. Training of data collectors is essential with emphasis on the risks to the participants and how to protect them, the importance of non-judgemental attitudes and provision of necessary support to participants.

Interviews with healthcare providers will be based on WHO's six building blocks framework[21] with a focus on the delivery of the focus SRH services during COVID-19; their perceptions on the roles and responsibilities of different parties to provide these services in the COVID-19 context; health system capacity to provide good quality of care for people during COVID-19; training needs, attitudes, biases about contraception, abortion, STI and SGBV in the context of COVID-19 and perceived psychosocial effects on men and women, their families and local communities.

The health availability and readiness assessment will be implemented in all the selected facilities using a cross-sectional survey design to highlight gaps or service delivery issues during COVID-19 and recovery

in service availability and readiness in the same health facilities. The assessment will also include a review of health facility records in family planning and contraception, STI care and treatment and care and support for women who have experienced SGBV to assess the availability, type and range of commodities offered. A follow-up assessment will be done at end-line. The follow-up survey plans to assess the recovery in service availability and readiness in the same health facilities. The research team will fill out the data extraction tool and the questionnaire with the assistance of the most knowledgeable person in the health facility about SRH services provided (senior healthcare provider and/or administrator) for each section of the questionnaire. The assessment will also include a questionnaire to the clients of the health facility.

The client questionnaire contains questions related to general characteristics of clients and their experience in seeking SRH services. All the selected health facilities will be required to collect these data from all clients seeking SRH services once every month throughout the study period, to capture trends in service availability and utilisation.

### Data analysis

Health facility assessment data will use the presence of the tracer items for the provision of these services, such as availability of guidelines, staff and essential commodities. Data will be entered into an electronic database using data entry programmes such as Epi data and CSPro. Study sites using electronic data collection may skip this step and for WHO-supported centres implementing the study of an online platform (OpenClinica) will be used. Descriptive analysis will be used to illustrate the basic characteristics of the different facilities, including the monthly number of clients, types of procedures provided, number of medical staff, stocks of drugs, etc.

Qualitative analysis will form the main analysis for this study, beginning while data collection is ongoing to assess progress and determine any problems. The audio-recorded data will be transcribed verbatim and de-identified by using ID numbers in place of names. Where required, transcriptions will be translated to English and back-translated and analysed using content analysis, according to the suggested steps by Elo and Kyngäs,[21] as illustrated in table 1. The WHO team based in Geneva, in conjunction with the study principal investigator and project team, will support and monitor the data analysis.

### Data management and access

Data management plans should include information about how data will be stored, including levels of protection, who will have access to the data and when it would be destroyed, how data will be transferred (if needed) securely. In countries where data protection legislation exists, protocols should specify that data would be handled in accordance with those policies.

**Table 1** Planned analytical procedures for content analysis

| Phase | Procedures |
| --- | --- |
| Preparation | Reading through verbatim transcriptions of the interviews several times to familiarise with the data, gain an understanding of what has been expressed, selecting the unit of analysis, deciding on the analysis of manifest content. |
| Organising | Open coding and creating categories, grouping codes under higher order headings, formulating a general description of the research topic through generating categories and subcategories as abstracting. |
| Reporting | Reporting the analysis process and the results through models, conceptual systems, conceptual map or categories and a story line. |

### Quantitative data

Data collectors will be trained on data collection, transmission, verification, storage and primary analysis to assess errors. Data can be collected electronically, or on paper.

### Qualitative data

The quality and trustworthiness of the qualitative data collected will be assured through triangulation, as we will conduct interviews with different parties, including health professionals, health workers and clients. The process of the study will be clearly documented to ensure the transparency and the rigour of the study.

### ETHICS AND DISSEMINATION

Scientific approval has been obtained from WHO research project review panel (RP2). Ethical approval for this study was also obtained from the WHO Ethics Review Committee (protocol ID CERC.0103). Each study site is required to obtain the necessary ethical and regulatory approvals that are required in the corresponding country. Ethical considerations of informed consent, voluntary participation, privacy and confidentiality, anonymity and compensation for incidental costs (the decision to offer compensation and its value will depend on the specific country context and respective local policies) will be respected and detailed in the informed consent process.

Given the sensitive nature of the data being collected, special care will be taken to ensure the 'do no harm' principle is respected. This will include making arrangements with locally available support services (medical, psychosocial, legal) where participants can be referred, training the research team to respond to and provide immediate emotional support to study participants or for situations where support services are not locally available or are inadequate.[22 23]

At country level, the results will be presented to policy makers, researchers, managers through policy briefs and

workshops in collaboration with the WHO office. The results will also be disseminated among the communities/participants through online platforms/text messages, which will be provided to the participants by the data collectors/health workers. In addition, the results will also be presented to the scientific and funding community in collaboration with WHO country and regional offices, by communication and manuscript publications in peer-reviewed international journals.

## DISCUSSION

This is a generic standardised protocol designed to be used across many different settings globally and to be useful as templates for use in health emergencies in the future. This will facilitate the comparison of results across regions and countries.

This study provides a unique opportunity to assess the availability, utilisation and readiness of the SRH services during the COVID-19 pandemic. The main lesson learnt will be the adaptations of the health system in pandemic situations and what can be done to ensure continuity of essential SRH services.

The mixed-methods approach and panel design with two data collection points enable a comprehensive analysis of the barriers to availability, utilisation and readiness of SRH services during COVID-19, as well as the postpandemic recovery as transmission is contained.

This study is well placed to advocate for the development and strengthening of policies and services that are responsive to the needs of women and girls during health emergencies, given the potential to exacerbate existing gender and social inequalities and increase the vulnerability of women and girls to preventable health risks. The use of a mainly qualitative approach places women's rights and needs at the centre of the debate. The inclusion of partners will enhance understanding of gender dynamics and support efforts towards identifying strategies to enhance positive male involvement and engagement in women's sexual reproductive health and rights.

**Author affiliations**
[1]Reproductive Health Unit, Institute for Research in Health Sciences, Ouagadougou, Burkina Faso
[2]Department of Sexual and Reproductive Health, World Health Organization, Geneve, Switzerland
[3]Department of Obstetrics and Gynaecology, Khon Kaen University, Nai Mueang, Thailand
[4]Department of Obstetrics and Gynaecology, State University of Campinas—UNICAMP, Campinas, Brazil
[5]Department of Obstetrics and Gynaecology, University of Campinas, Campinas, Brazil
[6]Department of population, family and Reproductive Health, University of Ghana, Legon, Ghana

**Collaborators** WHO HRP Social Science Research Team: Brazil (Luis Bahamondes, Jose Guilherme Cecatti); Burkina Faso (Eunice Chomi, Seni Kouanda); China (Kun Tang, Hanxiyue Zhang, Yifan Zhu, Yueping Guo, Ge Yang and Chunxiao Peng); Ghana (Deda Ogum Alangea, Kwasi Tropsey, Emefa Judith Modey); Pakistan (Rozina Karmaliani, Laila Ladak); Thailand (Pisake Lumbiganon, Jen Sothornwit); Kenya (Marleen Temmerman, Abdu Mohiddin, Ferdinand Okwaro); Italy (Massimo Mirandola, Maddalena Cordioli, Alessia Savoldi, Simone Garzon, Stefano Uccella, Ranieri Poli); UK (Nigel Sherriff, Alexandra Sawyer, Jorg Huber, Jaime Vera, Debbie Williams). WHO Secretariat (Moazzam Ali, Caron Kim, Hamsadvani Kuganantham, Igor Toskin, Vanessa Brizuela, Anna Thorson, Joy Jerop Chebet, Hugo Gamerro Abrego, Soe Soe Thwin, Armando Seuc).

**Contributors** MA and CK conceptualised and designed the protocol. All authors contributed to developing the detailed methodology. SK, ENC and MA drafted the first draft manuscript. SJ, LB, JGC, PL, ME, VB, HK and AHS edited the draft, provided critical inputs and approved the final written version for publication.

**Funding** This research has been supported by German Federal Ministry of Health (BMG) COVID-19 research and development funding to WHO. It has also been supported by the UNDP-UNICEF-WHO-World Bank Special Programme of Research, Development and Research Training in Human Reproduction (HRP); a co-sponsored programme executed by WHO. The adaptation and implementation of this protocol in the HRP collaborating centres in Brazil, Burkina Faso, China, Ghana, Italy, Kenya, Pakistan, Thailand and the UK was also funded by the same.

**Disclaimer** This report contains the collective views of an international group of experts and does not necessarily represent the decisions or the stated policy of WHO.

**Competing interests** None declared.

**Patient and public involvement** Patients and/or the public were not involved in the design, or conduct, or reporting, or dissemination plans of this research.

**Patient consent for publication** Not applicable.

**Provenance and peer review** Not commissioned; externally peer reviewed.

**ORCID iDs**
Caron Kim http://orcid.org/0000-0002-3574-4160
Luis Bahamondes http://orcid.org/0000-0002-7356-8428
Jose Guilherme Cecatti http://orcid.org/0000-0003-1285-8445
Vanessa Brizuela http://orcid.org/0000-0002-4860-0828
Moazzam Ali http://orcid.org/0000-0001-6949-8976

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
