## [Reviewer comments · BMJ Open]

ARTICLE DETAILS

TITLE (PROVISIONAL)	Health systems analysis and evaluation of the barriers to availability, utilization and readiness of sexual and reproductive health services in COVID-19 affected areas: A WHO mixed methods study protocol
AUTHORS	Kouanda, Seni; Nahyuha Chomi, Eunice; Kim, Caron; Jen, Sothornwit; Bahamondes, Luis; Cecatti, Jose; Lumbiganon, Pisake; Emefa, Modey; Brizuela, Vanessa; Kuganantham, Hamsadvani; Seuc, Armando; Ali, Moazzam

VERSION 1 – REVIEW

REVIEWER	Bolarinwa, Obasanjo University of KwaZulu-Natal, Department of Public Health Medicine
REVIEW RETURNED	06-Nov-2021

GENERAL COMMENTS	Title Health systems analysis and evaluation of the barriers to availability, utilization and readiness of sexual and reproductive health services in COVID-19 affected areas: A WHO mixed methods study protocol Abstract • Line 6 write COVID-19 in full• Line 13, I don't know why gender-based violence (GBV) is mixed up with SRH – This is a major flaw and would probably affect the whole protocol manuscript. To further buttress this point, GBV does not in any way appears in the study protocol title. Article summary • Line 43, point 2 – the information here is conflicting because the study is a mixed-method approach. You can't state this as a limitation Introduction • Please consider removing GBV in this protocol. Methods and analysis • Line 35, please kindly write the full meaning of WHO• Under the study setting and inclusion criteria, if the facilities' location is undecided now, at least the name of the country(ies) that will be involved should be stated here. The selected locations are? Other and general comments • The authors have prepared an excellent outstanding protocol for their study. While everything looks intelligently acceptable, there is a need to remove GBV from the study. The fact that it's added to WHO assessment doesn't mean everything must be accepted - verbatim.
--

REVIEWER	RamaRao, Saumya Population Council
REVIEW RETURNED	29-Nov-2021

GENERAL COMMENTS	This is an important protocol that will be used in different geographical settings. It will shed light on the extent to which key SRH services were affected by the COVID-19 pandemic. The following comments are provided for the authors to address:  1. It was not entirely clear as to which countries will participate in the study and it will be helpful to be explicit. The abstract and the body of the protocol indicate that it is "a standard generic protocol designed for use in different global settings." The list of authors includes those from Burkina Faso, Ghana, Brazil, Thailand and the WHO; and the acknowledgment section includes other country teams. Furthermore, it appears that ethical approval were provided by study sites. For all these reasons it will be helpful to know which countries will participate. Relatedly, it will be important to provide a rationale for site selection. 2. Please provide a rationale for why this set of 4 SRH services are included for study and not others? It appears that contraception, STI, abortion, and GBV are prioritized. Why not include ANC/delivery/PNC? Relatedly, there is inconsistency in which SRH service is being included. Page 10, lines 30-33 indicates ANC, delivery and postnatal care but elsewhere it does not include MCH (e.g., page 11 lines 15 to 19 do not include MCH; page 12 lines 14-15; Q5B in FGDs with women; Q5 IDI with HCPs)--please ensure internal consistency. 3. Related to point 2 above, please be explicit if you will be including HIV services (testing, counseling, prevention and treatment) within STI. 4. Clarify if the study facilities will be from public and private sectors. As well as the level of the facility; in some places, the protocol indicates that the facilities are hospitals and in the questionnaire seems to suggest those in cities. I strongly urge the authors to consider facilities lower than tertiary level facilities and not restrict to big cities to get a true picture of how SRH services are organized and delivered during an epidemic/pandemic. 5. The protocol needs to be consistent about the age bands for the study respondents. For example, in some parts of the protocol, the age bands are 18-49 but in other parts it suggests girls younger than 18 would be included (e.g., page 9, lines 47 to 51 talk about girls younger than 18 but line 27 indicates 18 and above); and mention is made of assent suggesting that minors could be included. It will be helpful if the authors were explicit about if they intend to include minors or not. Second, are unmarried women being included? The protections that will need to be put in place for minors and anybody else who might be stigmatized on the basis of demographics or the type of service being sought has to be thought out carefully. 6. It was good to see male partners being included in the protocol. Given that abortion and GBV data will be collected, and if perpetrators are men, it will key to ensure protections for the women. For this reason, the authors could consider collecting information from women and men from different households and are unrelated to each other, thereby offering a degree of protection to female respondents. 7. Include the procedures for pre-testing the questionnaires for local relevance and language. Please add information on the language of the interviews.
---

	8. Please include information on the training the data collectors will receive; their gender (e.g., male interviewers interviewing men; women interviewers interviewing women; women interviewers interviewing women and male respondents). 9. I presume the COVID-19 knowledge assessment will include vaccination, hesitancy, and vaccination status.
--	---

VERSION 1 – AUTHOR RESPONSE

Reviewer: 1

Dr. Obasanjo Bolarinwa, University of KwaZulu-Natal

Comments to the Author:

Title

Health systems analysis and evaluation of the barriers to availability, utilization and readiness of sexual and reproductive health services in COVID-19 affected areas: A WHO mixed methods study protocol

Abstract

- Line 6 write COVID-19 in full

This has been done, highlighted in yellow.

- Line 13, I don't know why gender-based violence (GBV) is mixed up with SRH – This is a major flaw and would probably affect the whole protocol manuscript. To further buttress this point, GBV does not in any way appears in the study protocol title.

Response: We thank the reviewer for this comment, but we respectfully disagree that GBV should be removed from the protocol. In line with the ICPD Programme of Action (2009)¹, Report of the Guttmacher-Lancet Commission on Sexual and Reproductive Health and Rights (2018)², and others^{3,4} sexual and gender-based violence are components of sexual and reproductive health. We have revised the wording in the protocol to reflect the focus on sexual violence, but also other forms of violence that may impact women's access to SRH services.

1 United Nations Population Fund (UNFPA), Report of the International Conference on Population and Development, Cairo, 5-13 September 1994, 1995, A/CONF.171/13/Rev.1, available at: <https://www.refworld.org/docid/4a54bc080.html>

2 Ann M Starrs, Alex C Ezech, Gary Barker, Alaka Basu, Jane T Bertrand, Robert Blum, Awa M Coll-Seck, Anand Grover, Laura Laski, Monica Roa, Zeba A Sathar, Lale Say, Gamal I Serour, Susheela Singh, Karin Stenberg, Marleen Temmerman, Ann Biddlecom, Anna Popinchalk, Cynthia Summers, Lori S Ashford. Accelerate progress—sexual and reproductive health and rights for all: report of the Guttmacher–Lancet Commission Lancet 2018; 391:2642-92 [https://www.thelancet.com/pdfs/journals/lancet/PIIS0140-6736\(18\)30293-9.pdf](https://www.thelancet.com/pdfs/journals/lancet/PIIS0140-6736(18)30293-9.pdf)

3 Muluken Dessalegn Muluneh, Virginia Stulz, Lyn Francis and Kingsley Agho. Gender based violence against women in Sub-Saharan Africa: a systematic review and meta-analysis of cross-sectional studies Int J Environ Res Public Health. 2020Feb;17(3):303 <https://pubmed.ncbi.nlm.nih.gov/32024080/>

4 Nancy Felipe Russo, Angela Pirlott. Gender-based violence, concepts, methods and findings. Ann N Y Acad Sci. 2006 Nov;1087:178-205 <https://pubmed.ncbi.nlm.nih.gov/17189506/>

Article summary

• Line 43, point 2 – the information here is conflicting because the study is a mixed-method approach. You can't state this as a limitation

Response: The comment is not very clear, because we have not mentioned that the mixed-method approach is a limitation, rather it is a strength. The point we were making was related to the recovery period, i.e., time between baseline and end line data collection. Given the differential and dynamic nature of COVID-19 in different countries, the current estimate of between 9-12 months may be different, meaning that the analysis for different countries may also be carried out at different times.

Introduction

• Please consider removing GBV in this protocol.

Response: We have replaced GBV with sexual and gender-based violence

Methods and analysis

• Line 35, please kindly write the full meaning of WHO

Response: This has been done, highlighted in yellow.

• Under the study setting and inclusion criteria, if the facilities' location is undecided now, at least the name of the country(ies) that will be involved should be stated here. The selected locations are?

Response: This is a generic protocol, which was not designed to be implemented in any specific country. It has been developed by researchers from WHO-World Bank Special Programme of Research, Development and Research Training in Human Reproduction (HRP) hub collaborating centres, where it has been adapted and is currently being implemented. Although ethical approval for the generic protocol has been provided by the WHO Ethics and Review Committee, individual countries (including the collaborating centres) wishing to adapt and implement it have been advised in the protocol to obtain ethical approval from their respective local bodies. Some of the HRP hub and collaborating centres that have adopted the protocol and are at varying stages of implementation include Burkina Faso, Kenya, Ghana, Pakistan, Brazil.

Other and general comments

• The authors have prepared an excellent outstanding protocol for their study. While everything looks intelligently acceptable, there is a need to remove GBV from the study. The fact that it's added to WHO assessment doesn't mean everything must be accepted -verbatim.

Response: We have replaced GBV with sexual and gender-based violence.

Reviewer: 2

Dr. Saumya RamaRao, Population Council

Comments to the Author:

This is an important protocol that will be used in different geographical settings. It will shed light on the extent to which key SRH services were affected by the COVID-19 pandemic. The following comments are provided for the authors to address:

1. It was not entirely clear as to which countries will participate in the study and it will be helpful to be explicit. The abstract and the body of the protocol indicate that it is "a standard generic protocol designed for use in different global settings." The list of authors includes those from Burkina Faso, Ghana, Brazil, Thailand and the WHO; and the acknowledgment section includes other country teams. Furthermore, it appears that ethical approval were provided by study sites. For all these reasons it will be helpful to know which countries will participate. Relatedly, it will be important to provide a rationale for site selection.

Response: This is a generic protocol, which was not designed to be implemented in any specific country. It has been developed by researchers from WHO-World Bank Special Programme of

Research, Development and Research Training in Human Reproduction (HRP) hub collaborating centres

Ethical approval for this protocol was provided by the WHO Ethics and Review Committee and individual countries (including the collaborating centres) have been advised (in the grey comment boxes) in the protocol to obtain ethical approval from their respective local bodies. Some of the HRP hub and collaborating centres that have adopted the protocol and are at varying stages of implementation include Burkina Faso, Kenya, Ghana, Pakistan, Brazil.

2. Please provide a rationale for why this set of 4 SRH services are included for study and not others?

It appears that contraception, STI, abortion, and GBV are prioritized.

Why not include ANC/delivery/PNC? Relatedly, there is inconsistency in which SRH service is being included. Page 10, lines 30-33 indicates ANC, delivery and postnatal care but elsewhere it does not include MCH (e.g., page 11 lines 15 to 19 do not include MCH; page 12 lines 14-15; Q5B in FGDs with women; Q5 IDI with HCPs)--please ensure internal consistency.

Response: This is a good question, but since we could not study all the components of SRHR, we decided on GBV, STI, contraception and safe abortion. We have developed another protocol on pregnancy and COVID19. These studies will be focus on ANC, delivery and PNC.

The inconsistency on page 10 has been corrected to ensure only the four focus areas are included in the protocol.

3. Related to point 2 above, please be explicit if you will be including HIV services (testing, counseling, prevention and treatment) within STI.

Response: The STI component includes HIV services. We have made this explicit in the protocol on page 4.

4. Clarify if the study facilities will be from public and private sectors. As well as the level of the facility; in some places, the protocol indicates that the facilities are hospitals and in the questionnaire seems to suggest those in cities. I strongly urge the authors to consider facilities lower than tertiary level facilities and not restrict to big cities to get a true picture of how SRH services are organized and delivered during an epidemic/pandemic.

Response: In the facility selection section, we have explained that a heterogenous sample of facilities will be selected to achieve a representative mix of facilities. Ideally this would include all levels of facilities within the health system, both public and private.

5. The protocol needs to be consistent about the age bands for the study respondents. For example, in some parts of the protocol, the age bands are 18-49 but in other parts it suggests girls younger than 18 would be included (e.g., page 9, lines 47 to 51 talk about girls younger than 18 but line 27 indicates 18 and above); and mention is made of assent suggesting that minors could be included. It will be helpful if the authors were explicit about if they intend to include minors or not. Second, are unmarried women being included? The protections that will need to be put in place for minors and anybody else who might be stigmatized on the basis of demographics or the type of service being sought has to be thought out carefully.

Response: The protocol specifies the 18-49 age-band. However, for some countries, the reproductive age band is from 15-49 years. Hence, some flexibility has been provided (in the grey comment box) for those countries, with provision to ensure ethical standards of assent and consent are followed. Vaccine hesitancy among women and health care providers has been addressed in another protocol under development.

6. It was good to see male partners being included in the protocol. Given that abortion and GBV data will be collected, and if perpetrators are men, it will key to ensure protections for the women. For this reason, the authors could consider collecting information from women and men from different

households and are unrelated to each other, thereby offering a degree of protection to female respondents.

Response: The protocol is designed to include the male partners to explore their influence on the access to SRH services of their partners and what role they play when their partners require SRH services, in the context of the pandemic. To protect the women, interviews for the women will be conducted separately from their partners. This has been explicitly mentioned on page 9.

7. Include the procedures for pre-testing the questionnaires for local relevance and language. Please add information on the language of the interviews.

Response: This was specified in the grey comment box on page 8 - After adaptation to specific country context all tools should be piloted before being used. We have added that the adaptation should include translation to the local language.

8. Please include information on the training the data collectors will receive; their gender (e.g., male interviewers interviewing men; women interviewers interviewing women; women interviewers interviewing women and male respondents).

This has been done (Page 10)

9. I presume the COVID-19 knowledge assessment will include vaccination, hesitancy, and vaccination status.

Response: The COVID-19 knowledge assessment was aimed understanding what people know about its transmission and linking this knowledge to people's risk perceptions and how this may have affected their health seeking behaviour. Vaccination, hesitancy and status while important are beyond the scope of this study.

VERSION 2 – REVIEW

REVIEWER	Bolarinwa, Obasanjo University of KwaZulu-Natal, Department of Public Health Medicine
REVIEW RETURNED	01-Feb-2022

GENERAL COMMENTS	Congratulations
-----------------

REVIEWER	RamaRao, Saumya Population Council
REVIEW RETURNED	07-Feb-2022

GENERAL COMMENTS	Thanks for addressing the comments provided--this version is cleaner and much better to read. Two issues remain: first, there continues to be slippage in referring to health facilities as "hospitals" and health facilities. Use the generic wording of "health facility" so that it can apply to all study situations. Second, I request the authors to carefully consider if interviewing husbands and wives from the same household is essential especially as I have noted before there can be issues when the husbands are the perpetrators of SGBV for which the wife is seeking care. The study can address gender dimensions of seeking care by interviewing husbands and wives from different households; by not interviewing husband-wife from the same household will operationalize the concept of "do no harm" and offer a measure of protection for those women who may be at risk just because they participated in the study.
---

VERSION 2 – AUTHOR RESPONSE

Reviewer: 2

Thanks for addressing the comments provided--this version is cleaner and much better to read. Two issues remain:

first, there continues to be slippage in referring to health facilities as "hospitals" and health facilities. Use the generic wording of "health facility" so that it can apply to all study situations.

Response: Thanks very much for your suggestion. We have replaced the term "hospital" with "health facility" in the manuscript text and for ease of reviewing have highlighted them in YELLOW.

Second, I request the authors to carefully consider if interviewing husbands and wives from the same household is essential especially as I have noted before there can be issues when the husbands are the perpetrators of SGBV for which the wife is seeking care. The study can address gender dimensions of seeking care by interviewing husbands and wives from different households; by not interviewing husband-wife from the same household will operationalize the concept of "do no harm" and offer a measure of protection for those women who may be at risk just because they participated in the study.

Response: Great suggestion. We have added the above-mentioned issue under the qualitative research section on page 4. It is highlighted in YELLOW.

It reads: "However, husbands and wives from different household will be interviewed to offer a measure of protection for those women who may be at risk of harm".